# A High-Efficient Reinforcement Learning Approach for Dexterous Manipulation

**DOI:** 10.3390/biomimetics8020264

**Published:** 2023-06-16

**Authors:** Jianhua Zhang, Xuanyi Zhou, Jinyu Zhou, Shiming Qiu, Guoyuan Liang, Shibo Cai, Guanjun Bao

**Affiliations:** 1College of Mechanical Engineering, Beijing University of Science and Technology, Beijing 100083, China; jhzhang@ustb.edu.cn; 2College of Mechanical Engineering, Zhejiang University of Technology, Hangzhou 310023, China; zhouxuanyi@zjut.edu.cn (X.Z.); 201906040306@zjut.edu.cn (S.Q.); 3Guangdong Provincial Key Laboratory of Robotics and Intelligent System, Shenzhen Institute of Advanced Technology, Chinese Academy of Sciences, Shenzhen 518055, China

**Keywords:** adaptive trajectory planning kernel, dynamic model, reinforcement learning, generative adversarial architecture

## Abstract

Robotic hands have the potential to perform complex tasks in unstructured environments owing to their bionic design, inspired by the most agile biological hand. However, the modeling, planning and control of dexterous hands remain unresolved, open challenges, resulting in the simple movements and relatively clumsy motions of current robotic end effectors. This paper proposed a dynamic model based on generative adversarial architecture to learn the state mode of the dexterous hand, reducing the model’s prediction error in long spans. An adaptive trajectory planning kernel was also developed to generate High-Value Area Trajectory (HVAT) data according to the control task and dynamic model, with adaptive trajectory adjustment achieved by changing the Levenberg–Marquardt (LM) coefficient and the linear searching coefficient. Furthermore, an improved Soft Actor–Critic (SAC) algorithm is designed by combining maximum entropy value iteration and HVAT value iteration. An experimental platform and simulation program were built to verify the proposed method with two manipulating tasks. The experimental results indicate that the proposed dexterous hand reinforcement learning algorithm has better training efficiency and requires fewer training samples to achieve quite satisfactory learning and control performance.

## 1. Introduction

Dexterous hands are considered to be the most complex and diverse end effector for robots. Compared to other end effectors, dexterous hands have more flexibility in grasping objects [1], especially irregular objects. However, performing excellent grasping and manipulation with dexterous hands remains a grand challenge.

Excellent grasping performance relies highly on interaction with the environment, where reinforcement learning algorithms have been applied for decades to solved countless sequential challenging problems [2]. In particular, deep reinforcement learning (DRL), a combination of reinforcement learning and deep neural networks, has achieved fruitful results in the field of deep learning [3,4] for robotics, such as robotic dogs, unmanned vehicles and humanoid robots [5]. High-level, domain-specific knowledge for robotic planning could be learned automatically. Moreover, the computational framework for robotic tasks and automation have been developed to facilitate the study and development of manipulation skills for robotic hand.

Transferring difficulties in robotics to reinforcement learning [6] has enabled robots to address previously intractable problems [7], and DRL has provided a powerful means of representing complex policies in high-dimensional environments [8,9]. A considerable number of scholars have reported their achievements in applications on real-world robots. For example, Tsurumine proposed a deep reinforcement learning network that combines the nature of smooth policy to enhance stability and efficiency for robotic cloth manipulation [10]. A path integration reinforcement learning algorithm was proposed to solve the control problem of tendon-wired robotic actuators without modeling the dexterous hand or environment, inspired by the human brain’s muscle synergy to obtain planning and control [11]. However, reinforcement learning in real-world robots is very time-consuming [12]. For instance, a model-free reinforcement learning algorithm applied to a three-finger dexterous hand system took seven hours of training time to complete various control tasks in the real world [13]. This slow training efficiency remains a stumbling obstacle to real-time robotic control. To accelerate the training speed, researchers have proposed various techniques. For example, a hierarchical planning method was proposed to achieve the movement of turning small balls by learning the best strategies for different levels in a decoupled manner [14]. Q-learning was applied to high-level discrete motions, and an improved path integration strategy was used to learn Dynamic Movement Primitives (DMPs) for low-level control. Adding a small amount of artificial demonstration data was found to be effective. In order to improve the efficiency of data use, a DPG-R algorithm was proposed based on the DDPG algorithm and Q-Learning algorithm. The efficiency of learning data samples was improved by separating the update frequency of the network from the environment [15]. Kumar designed a non-grasping five-finger dexterous hand by learning the local linear time-varying model of the dexterous hand and building a local trajectory planning controller for the model [16]. These controllers were first initialized with manual demonstration data. Moreover, the control strategy could be learned only through tactile and dexterous hand proprioceptive feedback without relying on the visual feedback of the object. Nagabandi proposed planning with a deep dynamics model (PDDM), a model-based reinforcement learning algorithm [17]. In contrast to the local linear time-varying model, deep neural networks were used to learn a global model of the system, and the learned model was used to plan motion based on the cross-entropy method (CEM). The algorithm was applied to control the multi-fingered dexterous hand to perform multiple tasks [18,19]. Andrychowicz from the OpenAI company proposed a technique that uses domain randomization to pre-train dexterous hands in the virtual world through reinforcement learning. The pre-trained model was then fine-tuned using transfer learning to operate on a physical robotic hand. To estimate object poses from visual information, multi-layer convolutional neural networks were commonly used [20]. Proximal policy optimization (PPO) was often employed to train dexterous hands in a virtual environment using thousands of different parameters [21]. However, one difficulty with PPO is that it may require a large amount of training data to achieve acceptable results.

To address this issue, Soft Actor–Critic (SAC) was proposed based on the maximum entropy theory. Compared to algorithms such as Deep Deterministic Policy Gradient (DDPG), SAC is more efficient in selecting initial network parameters. Additionally, due to the adoption of maximum entropy, SAC can achieve better control performance for different random seeds. Therefore, applying the SAC algorithm to the dexterous hand system to improve data learning efficiency is a promising approach. Overall, in applying domain randomization, transfer learning, and multi-layer convolutional neural networks, SAC can improve the efficiency of dexterous hand learning in both virtual and real-world environments.

In this paper, a dynamic model of the dexterous hand is proposed based on generative adversarial architecture to learn the dynamic changing principle of the state of the dexterous hand. The contributions of this paper are as follows:To address the problem of prediction error explosion over long spans, the state of the dexterous hand is learned through generative adversarial architecture with high prediction accuracy.An adaptive trajectory planning method is proposed. Moreover, an adaptive trajectory programming kernel is built to generate High Value Area Trajectory (HVAT) based on the dexterous dynamic model and object. A smooth distance loss function and a U-shaped loss function are designed to calculate the loss value of the dexterous hand system at the reference point.A new Actor–Critic-based reinforcement learning algorithm is proposed for the control of the dexterous hand.

This work enables the dexterous hand to explore high-value areas through a certain execution sequence, thereby improving the learning speed and performance of reinforcement learning algorithms.

## 2. Methodologies

The hybrid reinforcement learning strategy proposed in this paper is shown in Figure 1, which mainly consists of the SAC network, physical model of the dexterous hand and trajectory planning kernel. There are three networks in SAC, including CriticNet, ValueNet and ActorNet. Firstly, the system state obtained by the sensor in the manipulating environment is taken as the input of the SAC algorithm to the ${Actor}_{\phi }$ network. Then the output motion of the dexterous hand is generated by trajectory planning. As the dexterous hand executes the motion, the environment state is updated and employed to renew ${Actor}_{\phi }$ to generate the subsequent motion.

### 2.1. Generative Adversarial Architecture

In this study, the dexterous manipulation scenario is composed of a robotic hand and its operating target, which is a rotatable object such as a ball or blade. As shown in Figure 2, the state of the system includes variables such as the joint angle, joint velocity, joint torque, fingertip position, fingertip velocity, the position and velocity of the object, etc.

The state vector $s={\left[s_{0},\cdots ,s_{N}\right]}^{T}$ and output vector o are established for the kinematics and dynamics of the dexterous hand:(1)$$s^{\prime }=f\left(s,a\right)$$
where a is the input of the system.

Traditionally, the dynamic model of the system is established through analyzing the dynamic model. In some cases of collision and slide, the dynamic model is hard to establish accurately. In this paper, a deep neural network is used to learn the dynamic model of the dexterous hand. The state equation of the dynamic model is able to automatically be designed. Another advantage is that the ${Dyn}_{NN}$ is applied to obtain the changing principle of the system state. Therefore, when selecting the system state s, it only needs to satisfy that the state s satisfies the model of the Markov decision, instead of mapping the system state s into the observed value in traditional control methods.

The dynamic model of the system is obtained through the data-fitting method. Moreover, it can be used for model predictive control. Finally, a dynamic model of the system can be obtained:(2)$$s_{t+1}=f\left(s_{t},a_{t}\right)\approx {Dyn}_{NN}\left(s_{t},a_{t}\right)$$
where ${Dyn}_{NN}$ represents a neural network model.

An improved generative adversarial nets (GAN) architecture is proposed for building dexterous hand dynamic models. The improved generative adversarial architecture consists of two modules, a generative network module and adversarial network module ${Disc}_{\epsilon },$ which is represented by the network parameters ϑ and ϵ, respectively. The structure of ${Dyn}_{\vartheta }$ is shown in Figure 3.

The input of ${Dyn}_{\vartheta }$ is a two-dimensional vector $\left[\begin{array}{c} s_{t}\\ a_{t} \end{array}\right]$, composed of the state of the dexterous hand at *t*-moment and the input of *t*-moment system. The output $∆s_{t+1}$ of ${Dyn}_{\vartheta }$ is the difference between the predicted state $s_{t+1}$ of the dexterous hand at *t*-moment and the system state $s_{t}$ at *t*-moment.

When predicting the state of the dexterous hand system at the next moment, the current tensor ${\left[s_{t},a_{t}\right]}^{T}$ first passes through a one-dimensional convolution layer with a kernel size of 1, stride of 1 and out channels of 16. This convolutional layer is mainly used to extract the polarity characteristics of each element in ${\left[s_{t},a_{t}\right]}^{T},$ and the parameters of this convolutional layer are initialized to a normal distribution with a mean of 0 and variance of $\sqrt[{2}]{1/8}$. The tensor then passes through a linear and ReLu layer of 256 neurons for nonlinear mapping. Then the tensor continues to pass through a one-dimensional convolutional layer with a kernel size of 1, stride of 1 and out channels of 16. The parameters of this convolutional layer are initialized in the same way as those of the above convolutional layer, and are also used to extract the polar features of each element in the current layer. A basic microarchitecture of ${Dyn}_{\vartheta }$ is constructed by adjusting the polarity features of the extracted tensor and the number of nonlinear mapping levels. Theoretically, the more times the microarchitecture is reused, the more complex the dynamic model that can be learned by the ${Dyn}_{\vartheta }$ model. With the above structure, the final tensor passes through three mapping layers composed of linear and ReLu layers, and outputs the predicted value $∆s_{t+1}$.

The adversarial network module ${Dyn}_{\vartheta }$ structure is shown in Figure 4. The ${Disc}_{\epsilon }$ donates the dexterous hand system state and the predicted dexterous hand system state by True or False, respectively. If the input ${Disc}_{\epsilon }$ of the tensor is a true dexterous hand state, then the ${Disc}_{\epsilon }$ should be 1, or True. On contrary, if the tensor entered by the ${Disc}_{\epsilon }$ is predicted by ${Dyn}_{\vartheta },$ the output of the ${Disc}_{\epsilon }$ should be 0, or False. In practice, the ${Disc}_{\epsilon }$ outputs a value of $\left[0,1\right],$ and the larger the value, the more likely it is that the current input state is a real dexterous hand system state. ${Disc}_{\epsilon }$ also has the same structure of one-dimensional convolutional layers and nonlinear mapping as ${Dyn}_{\vartheta }$ for extracting polarity features.

In the dynamic model of a dexterous hand with a training model, the loss function of ${Dyn}_{\vartheta }$ is:(3)$$\begin{array}{c} L_{Dyn}(\vartheta )=E_{s_{t},a_{t},s_{t+1}\sim Buffer}[\chi \log\left(1-{Disc}_{\epsilon }\left({Dyn}_{\vartheta }\left(s_{t},a_{t}\right)+s_{t}\right)\right)\\ +\left(1-\chi \right)\left\|s_{t+1}-s_{t}-{Dyn}_{\vartheta }\left(s_{t},a_{t}\right)\right\|] \end{array}$$

The loss function of ${Disc}_{\epsilon }$ is:(4)$$L_{Disc}\left(\epsilon \right)=-E_{s_{t},a_{t}\sim Buffer}\left[\log\left({1-Disc}_{\epsilon }\left(s_{t}\right)\right)+\log\left({Disc}_{\epsilon }\left({Dyn}_{\vartheta }\left(s_{t},a_{t}\right)+s_{t}\right)\right)\right]$$

It can be concluded from the ${Dyn}_{\vartheta }$ network that the loss function consists of two parts, $\left\|s_{t+1}-s_{t}-{Dyn}_{\vartheta }\left(s_{t},a_{t}\right)\right\|$ and $\log\left(1-{Disc}_{\epsilon }\left({Dyn}_{\vartheta }\left(s_{t},a_{t}\right)+s_{t}\right)\right)$. The former is the prediction performance of the ${Dyn}_{\vartheta },$ which reflects the difference of the real states before and after the system under motion control input $a_{t}$. The latter can reflect the probability of the true state in the state of next moment ${\hat{s}}_{t+1}={Dyn}_{\vartheta }\left(s_{t},a_{t}\right)+s_{t}$ predicted by ${Disc}_{\epsilon }$ and ${Dyn}_{\vartheta }$. For the ${Disc}_{\epsilon }$ network, the loss function also consists of two parts, respectively, the $\log\left({1-Disc}_{\epsilon }\left(s_{t}\right)\right)$ and $\log\left({Disc}_{\epsilon }\left({Dyn}_{\vartheta }\left(s_{t},a_{t}\right)+s_{t}\right)\right)$. The former reflects the real state of the dexterous hand system for the possibility of True; and the latter reflects by the ${Dyn}_{\vartheta }$ prediction of the dexterous hand system status of the next moment ${\hat{s}}_{t+1}$ for the possibility of False. It is used for training ${Disc}_{\epsilon }$ by maximizing the loss function.

### 2.2. Theoretical Analysis of Trajectory Planning Algorithm

For a limited interval t=0:N, $\mu =\left\{a_{0},a_{1},\ldots ,a_{N-1}\right\}$ represents the control input sequence. A loss function $l\left(s_{t},a_{t}\right)$ satisfying concave function is used to calculate the system loss value at *t*-moment. The loss function belongs to a part of the control system. The loss value is calculated according to the target trajectory and the corresponding system trajectory. Then the cumulative loss value of the system in the interval k=i:N−1 can be expressed as:(5)$$J_{i}\left(s,\mu \right)=\sum_{i}^{N-1}l\left(s_{k},a_{k}\right)+l_{f}\left(s_{N}\right)$$

Afterwards, the optimal control problem can be expressed as finding an optimal trajectory sequence so that the loss accumulation value is a minimum within t=0:N:(6)$$\mu^{*}\left(s\right)=arg\underset{\mu }{\min}J_{0}\left(s,\mu \right)$$
where $\mu^{*}$ represents the optimal motion sequence.

The loss value function $V\left(s,i\right)=\underset{\mu_{i}}{\min}J_{i}\left(s,\mu_{i}\right),$ represents the cumulative value of system losses in the optimal motion subsequence $\mu_{i}^{*}=\left\{a_{i}^{*},a_{i+1}^{*},\ldots ,a_{N-1}^{*}\right\},$ and it is worth noting that the size of $V\left(s,i\right)$ only depends on the system state s. According to the memorylessness property, $V\left(s,i\right)$ can be obtained:(7)$$V\left(s,i\right)=\underset{a_{i}}{\min}\left[l\left(s_{i},a_{i}\right)+V\left(s_{i+1},i+1\right)\right]$$ It can also be obtained from $s_{i+1}=f(s_{i},a_{i})$:(8)$$V\left(s,i\right)=\underset{a_{i}}{\min}\left[l\left(s_{i},a_{i}\right)+V\left(f\left(s_{i},a_{i}\right),i+1\right)\right]$$
given an initial trajectory $\tau_{old}:\left\{s_{0},a_{0},\ldots ,s_{T}\right\}$ at t=0:N. In order to obtain the optimal sequence of motion, the control contains two parts, dynamic programming and optimal control. The method of dynamic programming is backpropagation and forward propagation. The optimal control under the linear system is discussed below, namely $f\left(s_{t},a_{t}\right)=F\left[\begin{array}{c} s_{t}\\ a_{t} \end{array}\right]$. In nonlinear systems, optimal control is obtained by iteration and local linearization.

Applying backpropagation, the Jacobian matrix and Hessian matrix of value function can be obtained at t=i:(9)$$V_{s}\left(i\right)=Q_{s}-Q_{a}Q_{aa}^{-1}Q_{as}$$
(10)$$V_{ss}\left(i\right)=Q_{ss}-Q_{sa}Q_{aa}^{-1}Q_{as}$$

$Q_{t},\;K_{t},\;k_{t},\;V_{s}\left(t\right)$ and $V_{ss}\left(t\right)$ in t=N:0 can be calculated according to the above equations, and the expression of the motion increment $\delta a_{t}$ with respect to the state change $\delta s_{t}$ at each time interval of t=0:N is obtained. The motion sequence is applied to the forward propagation and the motion $\tau_{new}$ is updated.

Applying forward propagation, the expressions of $K_{i}$ and $k_{i}$ at t=i are:(11)$$K_{i}=-Q_{aa}^{-1}Q_{as}=-{\left(l_{aa}+f_{a}^{T}\left(V_{ss}^{\prime }+\mu I\right)f_{a}+V_{s}^{\prime }f_{aa}\right)}^{-1}\left(l_{as}+f_{a}^{T}\left(V_{ss}^{\prime }+\mu I\right)f_{s}+V_{s}^{\prime }f_{as}\right)$$
(12)$$k_{i}=-Q_{aa}^{-1}Q_{a}=-{\left(l_{aa}+f_{a}^{T}\left(V_{ss}^{\prime }+\mu I\right)f_{a}+V_{s}^{\prime }f_{aa}\right)}^{-1}\left(l_{a}+f_{a}^{T}V_{s}^{\prime }\right)$$

The detail of the backpropagation and forward propagation are in the Appendix A. Finally, in the nonlinear system, the optimal control is calculated by successive iterations, and each iteration includes the above backward propagation and forward propagation.

### 2.3. Design of Loss Function

The dexterous hand system is a nonlinear system. In order to enhance the stability of the controller by reducing the change gradient of the loss function, a smooth distance function is designed to evaluate the state loss of the dexterous hand system:(13)$$l_{s}\left(s\right)=\sqrt{e^{2}+{\alpha_{s}}^{2}}-\alpha_{s}$$
where $\alpha_{s}$ represents the smoothness parameter, as shown in Figure 5, and $e=s-s_{goal}$ representants the system error. Choosing different loss function $\alpha_{s}$, the linearity will be regulated.

In this paper, $\alpha_{s}=0.2$ is adopted, and $l_{s}$ and $l_{ss}$ are defined as follows:(14)$$l_{s}=\frac{e}{\sqrt{e^{2}+{\alpha_{s}}^{2}}}$$
(15)$$l_{ss}={\left(e^{2}+{\alpha_{s}}^{2}\right)}^{-\frac{1}{2}}-e{\left(e^{2}+{\alpha_{s}}^{2}\right)}^{-\frac{3}{2}}$$

For the dexterous hand control input, it is necessary to ensure that the system control input has a small loss value within the allowable range. Moreover, after exceeding the system limit, the system loss value corresponding to the control input increases rapidly. In this paper, a U-shaped loss function is designed as follows:(16)$$l_{a}\left(a\right)={\alpha_{a}}^{2}\left(\cosh\frac{a}{\alpha_{a}}-1\right)$$
where $\alpha_{a}$ is a U-shaped coefficient.

When the control input is greater than 1 or less than −1, the loss value of the system feedback is too large. Figure 6 shows the image of the loss function for different $\alpha_{a}$. In this paper, $\alpha_{a}=0.3$ is adopted. The loss function, $l_{a}$ and $l_{aa}$ are expressed as:(17)$$l_{a}=\alpha_{a}\sinh\frac{a}{\alpha_{a}}$$
(18)$$l_{aa}=\cosh\frac{a}{\alpha_{a}}$$

### 2.4. Kernel Design of Adaptive Trajectory Planning

For nonlinear control systems, in order to obtain the optimal trajectory, there are three steps, as shown in Figure 7. Firstly, the reference point system model should be linearized, then the control feedback matrix *K* and *k* are obtained by backpropagation. As a result, the reference trajectory is obtained by forward propagation. An adaptive trajectory planning algorithm is built. By adjusting the linear search coefficient α and LM coefficient μ, the deviation can be controlled between the newly generated trajectory and the original reference trajectory. In the trajectory planning procedure, the existing control strategy is applied to generate an initial reference trajectory according to the initial system state. Then the program linearizes the reference points of the system at each moment of the reference trajectory with respect to ${Dyn}_{\vartheta }$ to obtain the local linear model, and obtains the Jacobian matrix and Hessian matrix of the loss function according to Equations (14), (15), (17) and (18).

Backpropagation is applied to calculate $V_{s}\left(t\right),\;V_{ss}\left(t\right),$ $Q_{t},\;K_{t},\;k_{t}$ and $V_{s}\left(N\right)=l_{s},\;V_{ss}\left(N\right)=l_{ss}$ from time *t* = N: 0. A linear search is carried out. The linear search coefficient is initialized as α=1, and the new motion sequence $A\{a_{0},a_{1},\ldots ,a_{N-1}\}$ [22] is calculated by $K_{t},\;k_{t}$ to obtain the new trajectory. If the loss value obtained along the new system trajectory is less than that of the original trajectory, it means that the program accepts this iteration. However, if the loss value obtained along the trajectory is greater than that corresponding to the original system trajectory under the current coefficient, the change value is reduced by dividing the linear search coefficient α by 1.1. Moreover, the motion sequence is recalculated and the loss value is compared.

Then, if the linear search coefficient α has reached the minimum value, the LM coefficient μ is increased and $K_{t},k_{t}$ are recalculated to impose greater constraints on the deviation from the original trajectory. Afterwards, the linear search link is re-activated. The process is repeated until a new trajectory is adopted. By adjusting the linear search coefficient α and LM coefficient μ, the adaptive planning trajectory is generated.

### 2.5. Improved SAC Algorithm

In this paper, an improved SAC algorithm is proposed to generate the motion of the dexterous hand, including ${Value}_{\psi }$ and ${TargetValue}_{\overset{-}{\psi }}\left(s_{t}\right)$ networks to evaluate the value of motions in the current state and ${Actor}_{\phi }$ network. The improved Bellman equation is obtained by incorporating the entropy of the control policy action distribution under the current system state s_t.
(19)$$Q\left(s_{t},a_{t}\right)∶=r\left(s_{t},a_{t}\right)+\gamma E_{s_{t+1}\sim p\left(s_{t+1}|s_{t},a_{t}\right)}\left[V\left(s_{t+1}\right)\right]$$

The data cache DataBuffer has an area for HVAT to store HVAT data generated through trajectory planning. In addition, this part of the data is generated by trajectory planning in the fitted dynamic model of the dexterous hand rather than obtained in the actual dexterous hand environment by the control strategy π. The loss functions of ${Critic}_{\theta }$ and ${Value}_{\psi }$ under HVAT are obtained as follows:(20)$$J_{Critic}^{HVAT}\left(\theta \right)=E_{\left(s_{t},a_{t},{EUCTN}_{t}\right)\sim \tau \_Buffer}\left[\frac{{conf}_{t}}{2}{\left({Critic}_{\theta }\left(s_{t},a_{t}\right)-\hat{Q}\left(s_{t},a_{t}\right)\right)}^{2}\right]$$
where ${conf}_{t}$ is the confidence value predicted by ${Dyn}_{\vartheta }$ under $s_{t},a_{t},$ $conf=\frac{1}{{EUCTN}_{t}^{{Dyn}_{\vartheta }}}$. EUCTN represents the cognitive uncertainty and can be written as:(21)$${EUCTN}_{t}^{{Dyn}_{\vartheta }}\approx \frac{\sum_{i}^{ES}{\left({∆s_{t+1}}_{i}-\bar{∆s_{t+1}}\right)}^{2}}{ES}$$
where ES=3 is the model number of the ensemble, $∆s_{t+1}$ is the output of the ${Dyn}_{\vartheta }$ and $\bar{∆s_{t+1}}$ is the average predicted value.

The loss function of ${Value}_{\psi }$ can be written as:(22)$$J_{Value}^{HVAT}\left(\psi \right)=E_{s_{t},a_{t}\sim HVAT\_Buffer}[\frac{1}{2}{\left({Value}_{\psi }\left(s_{t}\right)-\left(\frac{1}{2}\left(E_{{a\prime }_{t}\sim \pi }\left[{Critic}_{\theta }\left(s_{t},{a\prime }_{t}\right)\right]+{Critic}_{\theta }\left(s_{t},a_{t}\right)\right)-\log\pi \left({a\prime }_{t}|s_{t}\right)\right)\right)}^{2}]$$
where $a_{t}$ is sampled in the HVAT area. So the evaluation scope of ${Value}_{\psi }$ includes the motion distribution space generated by trajectory planning.

At this point, the loss function gradient for ${Critic}_{\theta }$ and ${Value}_{\psi }$ is defined as:(23)$${\hat{\nabla }}_{\theta }J_{Critic}^{HVAT}\left(\theta \right)={conf}_{t}\nabla_{\theta }{Critic}_{\theta }\left(s_{t},a_{t}\right)\left({Critic}_{\theta }\left(s_{t},a_{t}\right)-r\left(s_{t},a_{t}\right)-\gamma {TargetValue}_{\overset{-}{\psi }}\left(s_{t+1}\right)\right)$$
(24)$${\hat{\nabla }}_{\psi }J_{Value}^{HVAT}\left(\psi \right)=\nabla_{\psi }{Value}_{\psi }\left(s_{t}\right)\left({Value}_{\psi }\left(s_{t}\right)-\left(\frac{1}{2}\left(E_{{a\prime }_{t}\sim \pi }\left[{Critic}_{\theta }\left(s_{t},{a\prime }_{t}\right)\right]+{Critic}_{\theta }\left(s_{t},a_{t}\right)\right)-\log\pi \left({a\prime }_{t}|s_{t}\right)\right)\right)$$

Different from the direct generation of the trajectory by the control strategy, this method makes the high-value state region updated preferentially in the value iteration; thus, it indirectly accelerates the learning of the control strategy and avoids the direct influence of the low-value trajectory on the updating strategy of control.

#### Iteration of Control Policies

To maximize the entropy of actor–critic, the updating rule for the control policy π is defined as:(25)$$\pi_{new}=\arg\underset{\phi }{\min}D_{KL}\left(\pi \left(\cdot |s_{t}\right)\|\frac{exp\left(Q^{\pi_{old}}\left(s_{t},\cdot \right)\right)}{Z^{\pi_{old}}\left(s_{t}\right)}\right)$$
where $Z^{\pi_{old}}(s_{t})$ is the sum of all $exp(Q^{\pi_{old}}(s_{t},\cdot ))$.

Since Critic is used to denote Q, the loss function for $\pi_{\phi }$ can be expressed as follows:(26)$$J_{\pi }\left(\phi \right)=E_{s_{t}\sim Buffer}\left[D_{KL}\left(\pi_{\phi }\left(s_{t}\right)\|\frac{exp\left({Critic}_{\theta }\left(s_{t},\cdot \right)\right)}{Z_{\theta }\left(s_{t}\right)}\right)\right]$$

Meanwhile, ${Actor}_{\phi }\left(\epsilon_{t};s_{t}\right)$ neural network is applied for reparameterization:(27)$$a_{t}={Actor}_{\phi }\left(\epsilon_{t};s_{t}\right)$$
where $\epsilon_{t}$ is the noise vector sampled from the Gaussian distribution.

Therefore, Equation (24) can be rewritten as:(28)$$J_{\pi }\left(\phi \right)=E_{s_{t}\sim Buffer,\epsilon_{t}\sim N}\left[\log\pi_{\phi }\left({Actor}_{\phi }\left(\epsilon_{t};s_{t}\right)|s_{t}\right)-{Critic}_{\theta }\left(s_{t},{Actor}_{\phi }\left(\epsilon_{t};s_{t}\right)\right)\right]$$

It is worth noting that ${Actor}_{\phi }$ identifies a unique control strategy $\pi_{\phi }.$ Its gradient can be approximated as:(29)$${\hat{\nabla }}_{\phi }J_{\pi }\left(\phi \right)=\nabla_{\phi }\log\pi_{\phi }\left(a_{t}|s_{t}\right)+\left(\nabla_{a_{t}}\log\pi_{\phi }\left(a_{t}|s_{t}\right)-\nabla_{a_{t}}{Critic}_{\theta }\left(s_{t},a_{t}\right)\right)\nabla_{\phi }{Actor}_{\phi }\left(\epsilon_{t};s_{t}\right)$$

Finally, the parameters of the ${Actor}_{\phi }$ network are updated using gradient descent:(30)$$\phi \leftarrow \phi -\lambda_{\pi }{\hat{\nabla }}_{\phi }J_{\pi }\left(\phi \right)$$
where $\lambda_{\pi }$ is the learning rate of the control strategy π.

## 3. Experiments

### 3.1. Model-Fitting Experiments for Dynamic Processes

The dynamic model-fitting experiment of the dexterous hand is performed in a customized two-fingered hand simulation environment, as shown in Figure 8.

The hyperparameters used in this experiment are shown in Table 1 and Table 2. The cycle in the experimental simulation environment is 0.02 s, and the time step used in each simulation is 500.

We designed two types of prediction errors as performance indexes for the dynamic model of the dexterous hand based on the number of predicted steps; that is, short-span and long-span prediction errors. These indexes are used to analyze the variation principle of prediction errors and the phenomenon of error explosion with an increase in learning times. To do so, we initialized the dexterous hand system status and used the same control strategy for both types of prediction errors. The short-span prediction error is calculated based on the dynamic model and is obtained by predicting the state change of the dexterous hand system according to the reference trajectory in the next moment. The calculation for the short-span prediction error is:(31)$$\delta =mean\left(E_{s_{t},a_{t}\sim \tau }\left[\sum_{i=0}^{T-1}\left|s_{i+1}-{\hat{s}}_{i+1}\right|\right]\right)$$
where ${\hat{s}}_{i+1}=s_{i}-{Dyn}_{\vartheta }\left(s_{i},a_{i}\right).$

The long-span prediction error is used to predict the complete state trajectory of the dexterous hand system based on its initial state and motion sequence, and the prediction error is calculated by comparing it with the reference state trajectory. Unlike the short-span prediction error, the long-span prediction error can better reflect the model’s ability to learn the overall trend of the system state. The calculation for the long-span prediction error is:(32)$$\delta =mean\left(E_{s_{0},\left(s_{t},a_{t}\right)\sim \tau }\left[\sum_{i=0}^{T-1}\left|s_{i+1}-{\hat{s}}_{i+1}\right|\right]\right)$$
(33)$$\begin{array}{l} {\hat{s}}_{i+1}=s_{i}+{Dyn}_{\vartheta }({\hat{s}}_{i},a_{i})\\ {\hat{s}}_{i}={\hat{s}}_{i-1}+{Dyn}_{\vartheta }\left({\hat{s}}_{i-1},a_{i-1}\right)\\ {\hat{s}}_{0}=s_{0} \end{array}$$

Then, the fixed state of the transfer dataset was utilized to train the dynamic model of the dexterous hand. Initially, the SAC algorithm was employed to generate motion sequence, and the control strategy was updated according to the control task to obtain the system state transfer dataset, which includes 10,000 system state transfer tuples. The dataset was then used to train the two models the same number of times. After each training, the prediction performance was tested five times, and the average value was computed to obtain the model’s prediction error. Moreover, after each training, the dynamic model’s two different span prediction errors, with and without the generative adversarial architecture, were recorded, and the error trend chart was obtained, as shown in Figure 9. In the figure, the horizontal direction represents an increase in the predicted time, the vertical direction represents the state size of the dexterous hand system and the red curve represents the real system state, while the blue curve represents the system state predicted by the dynamic model.

Figure 9a illustrates the trend of short- and long-span prediction errors in the dynamic model with increasing training times. The red curve indicates that the generative adversarial architecture was adopted, while the blue curve indicates that the architecture was not adopted. The smaller the value in the figure, the higher the prediction accuracy. The lower part of the figure is divided into four subgraphs, respectively, based on the dynamic model of generative adversarial architecture in training, to reach 50(I), 100(II), 150(III), 200 times(IV) of full long-span prediction, forecasting the state change curve and the true state graph.

As can be seen in Figure 9, with the increase in the number of dynamic model trainings, the overall prediction accuracy of the model for the state change of the environment was gradually improved. The model without the generative adversarial architecture was gradually stabilized after 120 trainings on the short-span prediction error, which is slightly better than that using the generative adversarial architecture mode. The short-span prediction error using the generative adversarial architecture model shows an upward trend at the initial stage of training, and eventually falls back to a smaller value, which is due to the fact that the generative adversarial architecture was used for training. Because the loss function of ${Dyn}_{\vartheta }$ in Equation (3) contains adversarial loss items, the error value in the graph experienced an upward and then a downward trend during the training progress.

Regarding long-span prediction performance, the use of the generative adversarial architecture significantly reduced the prediction error compared to the model without the architecture. The prediction error of the former eventually stabilized at the same order of magnitude as the short-span prediction error, while the latter exhibits the phenomenon of error explosion. These results demonstrate that the dynamic model using the generative adversarial architecture can capture the overall trend of the system state change without significant loss in short-span prediction performance. Thus, when the control strategy remains unchanged, the prediction performance of the model for long spans can be significantly improved, thereby enhancing the model’s overall prediction accuracy.

### 3.2. Adaptive Trajectory Planning Experiments of the Dexterous Hand

In order to verify the performance of the trajectory planning kernel, experiments with the dexterous hand were conducted. The experimental hyperparameters are listed in Table 3:

In order to reduce the computational complexity, the nimble fingers space was selected as an important condition. A total of 100 rounds of dexterous hands’ state transfer data were collected firstly, with a total number of 100 steps for each round. Then, the dynamic model was fitted. Moreover, the dynamic model and the trajectory planning kernel were used for model predictive control. Then, 10 steps were predicted forward at each time, and the spatial coordinates of the fingertips of dexterous hands were randomly generated as target positions for trajectory planning experiments.

Figure 10 shows the fingertip trajectory of the dexterous hand under a total of 400 times of dynamic model training and under different training times with the dynamic model, where green and red represent finger 1 and finger 2, respectively, the dot represents the initial coordinate of the fingertip, the asterisk is the target coordinate of the fingertip and the curve is the actual trajectory of the corresponding fingertip. Figure 9 selected the dynamic model under different training times using adaptive kernel trajectory planning; in the fingertip trajectory graph, we can see that with the increase in dynamic model training, at the same time and steps, the tip of the finger position with respect to the target movement shows a trend of gradually ascending, and the trajectory planning effect also was improved.

According to the experiment, in the early stage of the dynamic model training, due to the large cognitive deviation of the model in the state change of the environment, the dynamic model error occurred in the fingertip trajectory planning near the start point, which led the trajectory planning into local minimum, wrong direction and other problems, resulting in the fingertip’s mismatch with the desired position. Although the method of model predictive control can reduce the trajectory error caused by the dynamic model to a certain extent, the performance of the dynamic model had a major impact on the trajectory planning within a certain number of control steps. In the later stages of dynamic model training, a high-value fingertip trajectory, namely HVAT data, could be obtained through adaptive trajectory planning due to the improved accuracy of the model.

### 3.3. Controlling Experiments of the Dexterous Hand

The HVAT data are obtained via the above method and the dexterous hand control experiment was carried out by combining the SAC algorithm. The pseudo-code of the specific algorithm in the experiment is shown in Table 4.

The experimental platform is shown in Figure 11, which includes a two-fingered hand and its sensing and controlling units, a camera and the ROS program.

The experimental results of fingertip controlling are shown in Figure 12, in which the algorithm proposed in this paper (the green one) is compared with the model-free reinforcement learning algorithms SAC [22], PPO [23], DDPG [24], Policy Gradient [24,25,26] and Actor–Critic [25]. A total of 250 sets of training, which contain 5000 simulation time units, are adopted. From Figure 12, it can be concluded that the exploration ability of the model-free reinforcement learning algorithm is improved. By applying the maximum entropy control strategy, the experimental curve can achieve the maximum reward in the shortest time. Therefore, the overall learning speed of the algorithm is faster than that of the others.

For the task of rotating the pointer, the proposed control algorithm achieved the expected motion to a specified angle locally on the experimental platform by learning the relationship between the dexterous hand movements and the state changes of the pointer. Figure 13 shows the captured image by the camera of the experimental platform during the experiment, in which the blue numbers at the top of the image represent the randomly generated desired angle and the white numbers represent the current angle of the pointer. It can be seen from the photos that the dexterous hand can change the angle of the pointer by touching it. Figure 14 shows the curve of the angle changes of the pointer in this experiment.

Meanwhile, comparative experiments were conducted on different reinforcement learning algorithms for the same experimental environment, as shown in Table 5, which indicates the number of sample collection times required by different algorithms to achieve the same reward value. Compared with the others, it can be found that the control algorithm proposed in this paper requires fewer data samples to achieve the same control effect; i.e., the average reward value is in the same range in each round of the control experiment. It should be noted that the reward size of each round is related to the duration of the control steps. The control time steps of each round are set to be 5000.

The experimental results once again proved that the model-based reinforcement learning algorithm proposed in this paper for dexterous hand control can significantly improve the efficiency of training and enhance the learning speed compared to other model-free reinforcement learning algorithms.

## 4. Conclusions

Controlling both dexterous hand motion and in-hand manipulation in an unstructured environment with multiple tasks remains a great challenge. To address this issue, one feasible solution is to allow dexterous hands to interact with the environment and learn to achieve control goals through interaction, similar to that of humans.

In this paper, we proposed a dynamic model of dexterous hands based on generative adversarial architecture, which is capable of learning the dynamic change principle of the dexterous hand system state. To solve the problem of prediction error explosion over long spans, the generative adversarial architecture was employed to learn the entire change rule of the dexterous hand system state while maintaining local prediction accuracy. Then, an iterative optimal control theory of trajectory planning was proposed. An adaptive trajectory planning kernel was built based on this theory to generate the HVAT trajectory according to the dynamic model of the dexterous hand. The kernel can adjust the linear search coefficient α and LM coefficient μ to constrain the change of trajectory, so as to achieve the purpose of self-adaptation. Subsequently, a U-shaped loss function was designed to calculate the loss value of the dexterous hand system at the reference point. Finally, the SAC algorithm was improved and a new Actor–Critic-based reinforcement learning algorithm was proposed for the end-to-end control of dexterous hands, including two critic networks for evaluating the value of motions in the current state. A ${Value}_{\psi }$ and ${TargetValue}_{\overset{-}{\psi }}\left(s_{t}\right)$ network was used to assess the value of the current state, and an ${Actor}_{\phi }$ network was used to generate motions. For random data in the cache, the algorithm adopts the update strategy of the maximum entropy framework, while for HVAT data in the cache, the algorithm adopted the hybrid update strategy.

The proposed algorithm can be used to enable dexterous hands to explore in high-value areas through a certain execution sequence, thereby improving the learning speed and performance of reinforcement learning algorithms. The dexterous hand was enabled to explore high-value areas through a specific execution sequence, thereby improving learning speed and performance. The above contributions provide a robust framework for end-to-end control of dexterous hands in unstructured environments with multitasking capabilities.

## Figures and Tables

**Figure 1 biomimetics-08-00264-f001:**
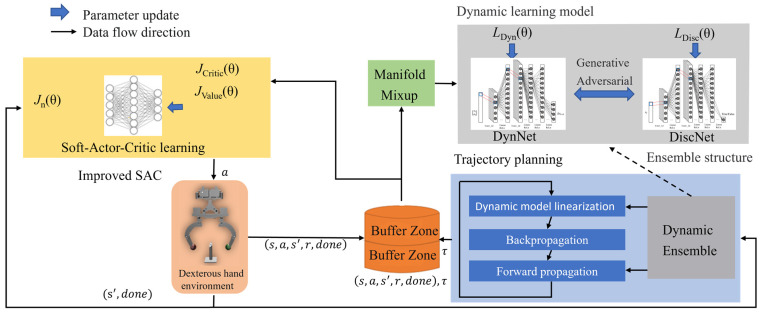
Schematic diagram of the proposed method.

**Figure 2 biomimetics-08-00264-f002:**
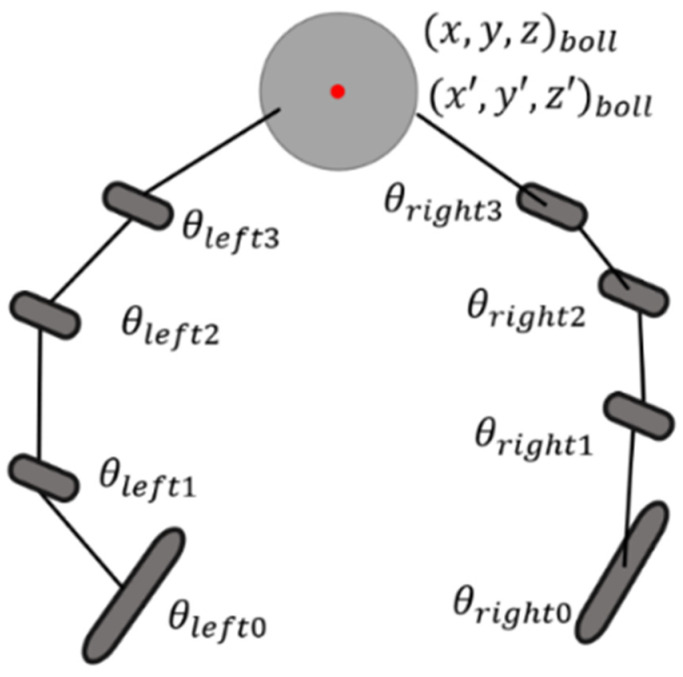
Dynamic model of the dexterous hand.

**Figure 3 biomimetics-08-00264-f003:**
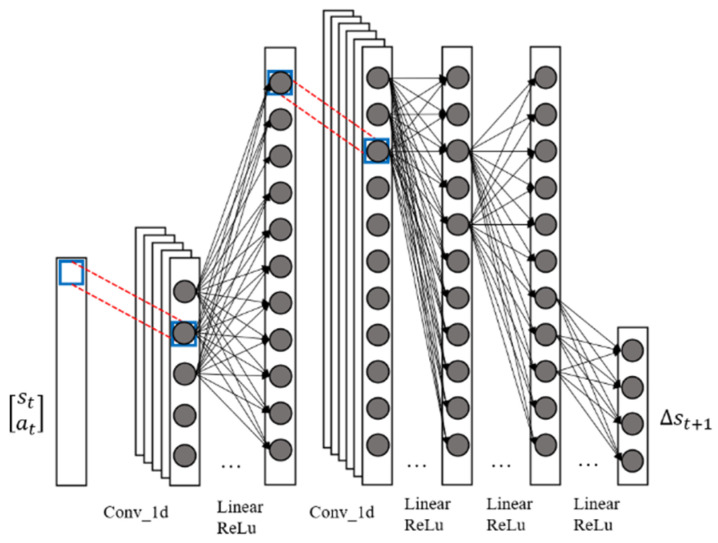
Generative network diagram for the dexterous hand.

**Figure 4 biomimetics-08-00264-f004:**
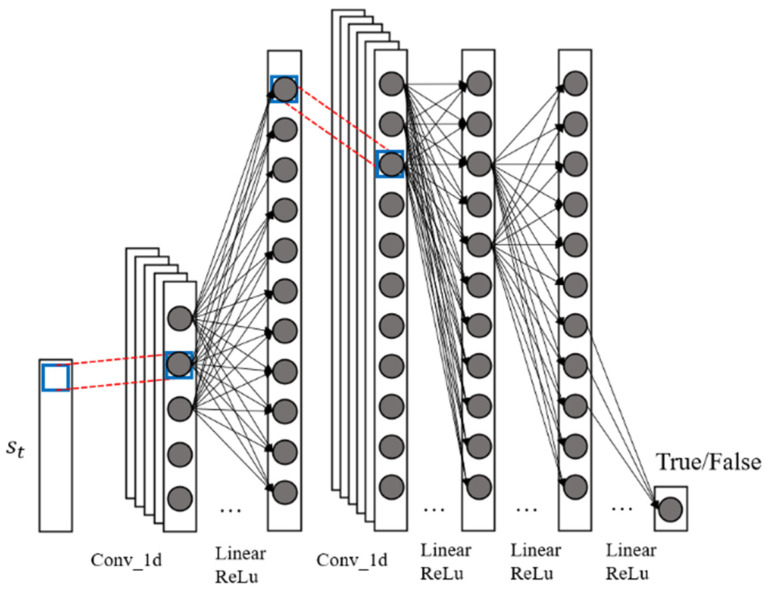
Adversarial network diagram of the dexterous hand.

**Figure 5 biomimetics-08-00264-f005:**
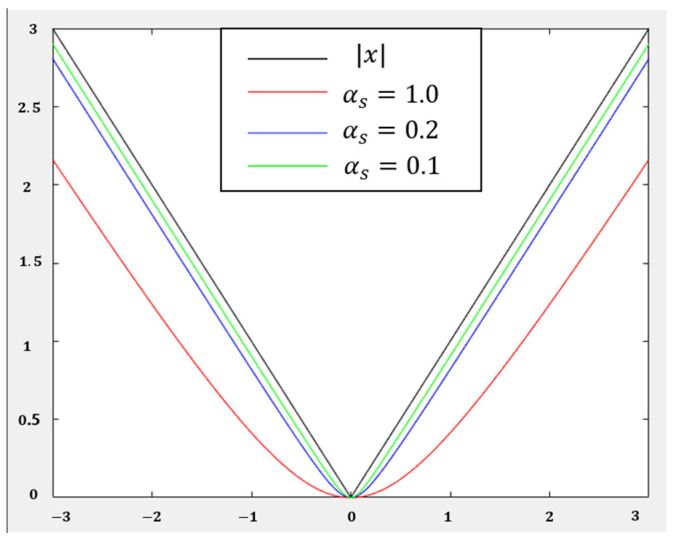
Loss function for different $\alpha_{s}$.

**Figure 6 biomimetics-08-00264-f006:**
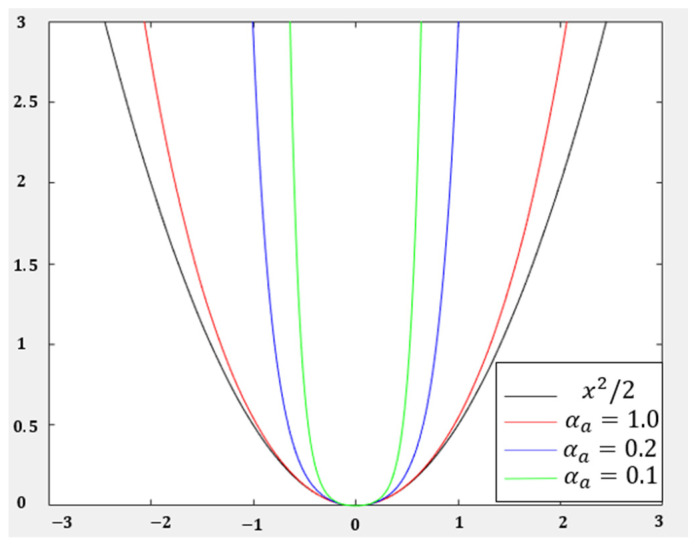
U-shaped loss function for different $\alpha_{a}$.

**Figure 7 biomimetics-08-00264-f007:**
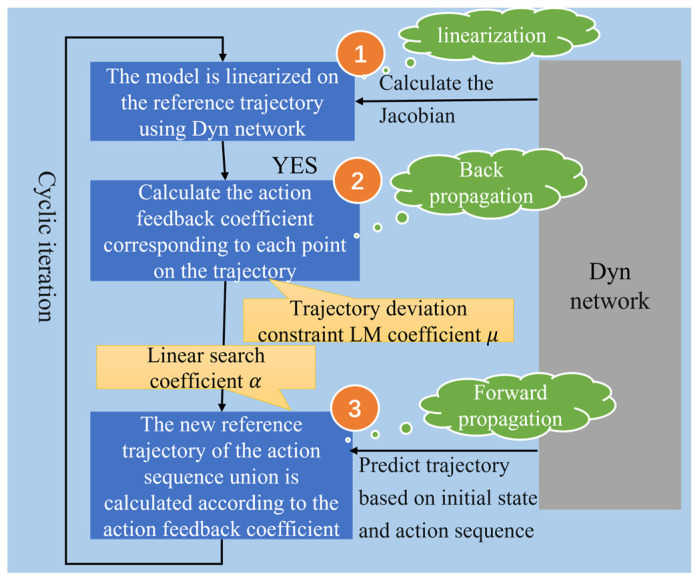
Steps for adaptive trajectory planning.

**Figure 8 biomimetics-08-00264-f008:**
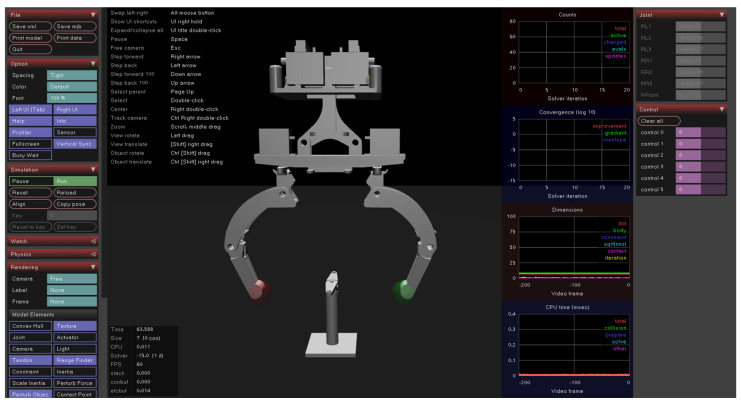
Simulation environment of a two-fingered hand.

**Figure 9 biomimetics-08-00264-f009:**
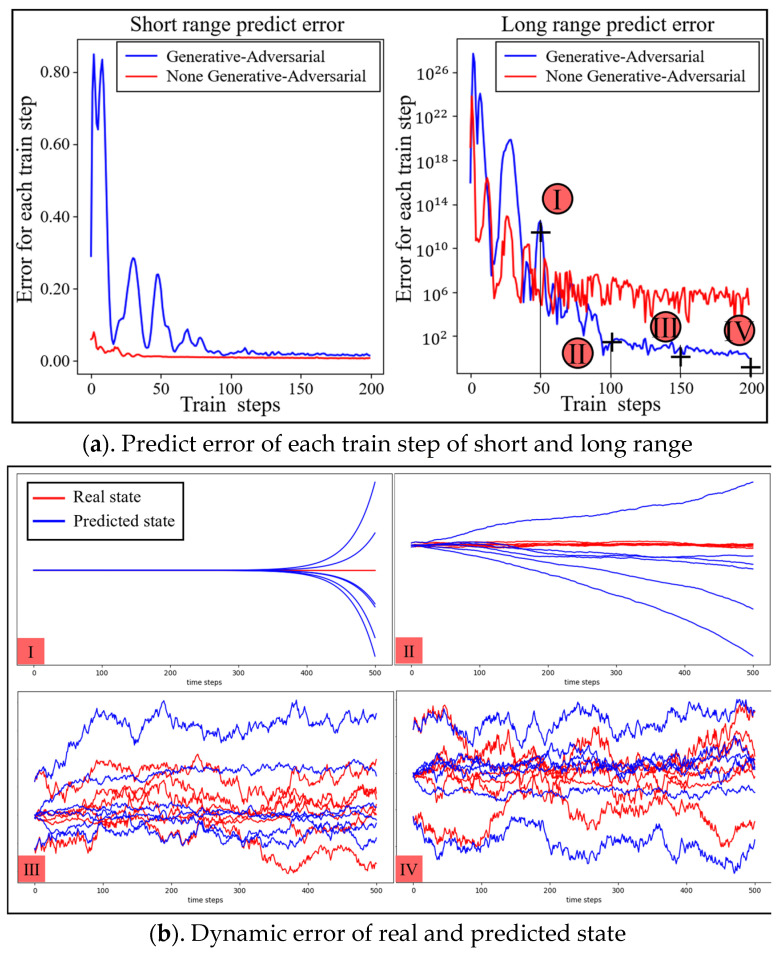
Evolution of the dynamic model’s prediction error.

**Figure 10 biomimetics-08-00264-f010:**
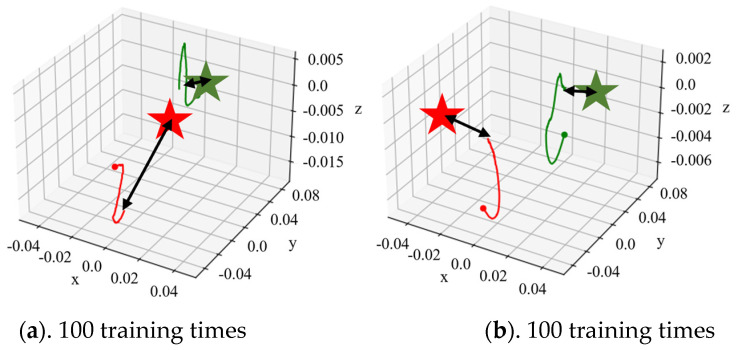
The trajectory of the dexterous hand.

**Figure 11 biomimetics-08-00264-f011:**
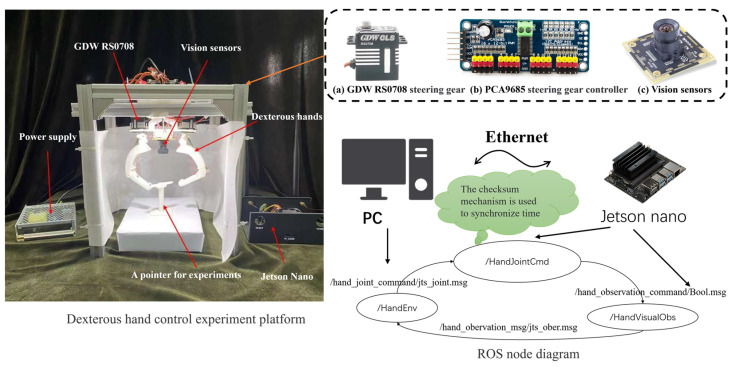
Scheme of the experimental platform.

**Figure 12 biomimetics-08-00264-f012:**
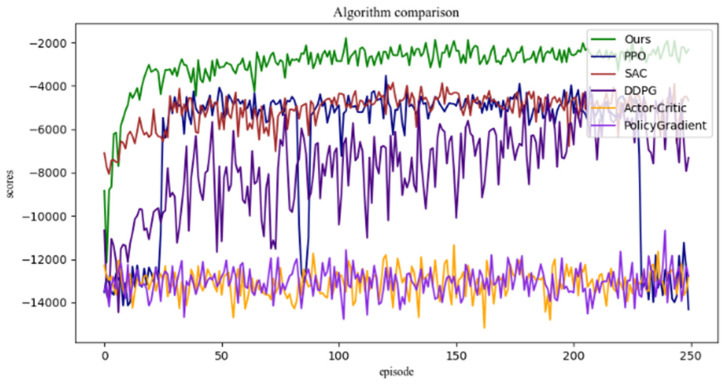
Fingertip controlling experimental results.

**Figure 13 biomimetics-08-00264-f013:**
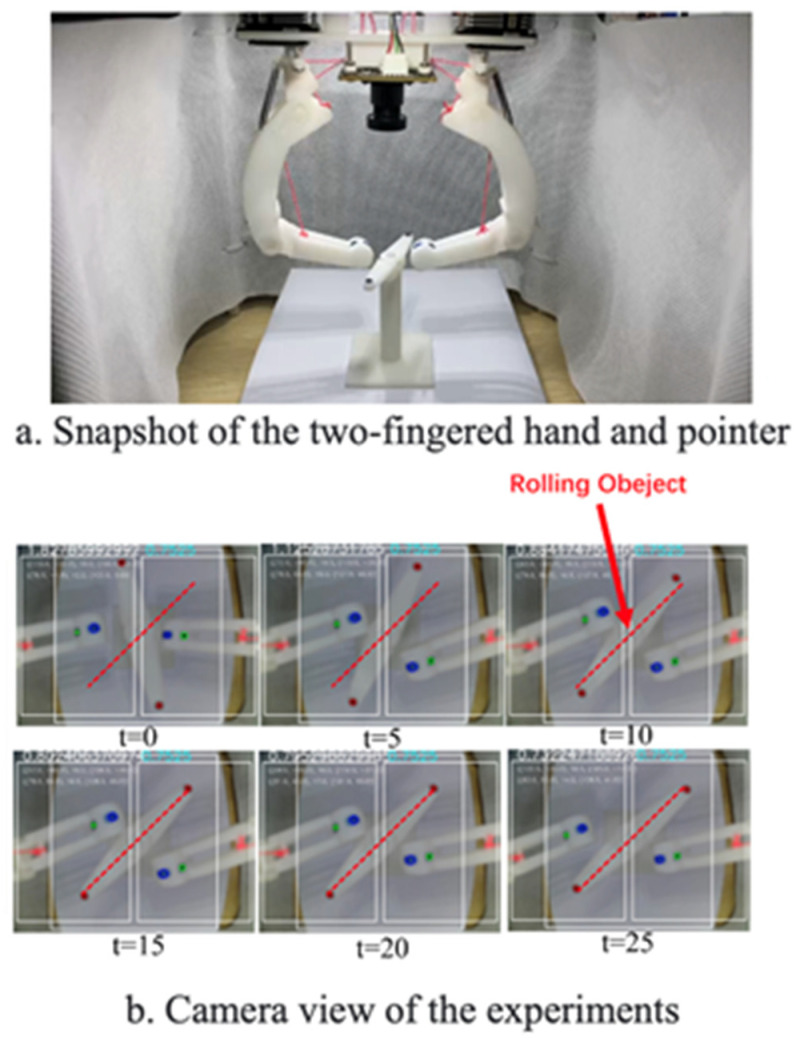
Rotation pointer experiments.

**Figure 14 biomimetics-08-00264-f014:**
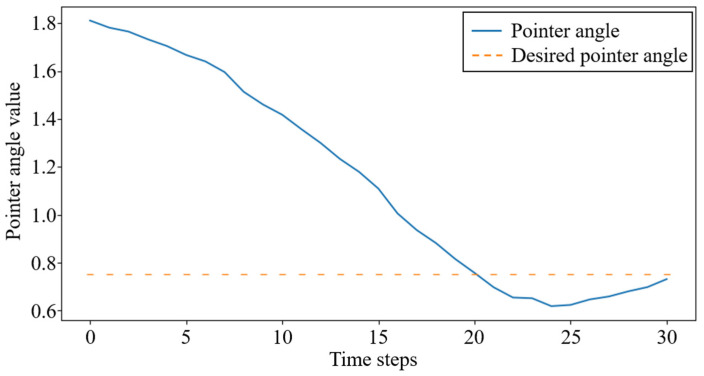
Convergence process of the pointer orientation.

**Table 1 biomimetics-08-00264-t001:** Parameters of the improved SAC algorithm.

Parameter	Symbol	Value
Learning rate	lr	0.0003
Hidden layer	deep	2
Number of monolayer neurons		256
The entropy coefficient	tmp	10
Discount factor	γ	0.99
Batch size		256

**Table 2 biomimetics-08-00264-t002:** Parameters of the dynamic model of the dexterous hand.

Parameter	Symbol	Value
Learning rate	lr	0.0001
Number of monolayer neurons		256
Number of polar layers		2
Adversarial coefficient	χ	0.4
Weight decay		0.0001

**Table 3 biomimetics-08-00264-t003:** Kernel hyperparameters of adaptive trajectory planning.

Parameter	Symbol	Value
Initial LM coefficient	$\mu_{0}$	0.1
Maximum LM coefficient	MAXμ	10
Gain coefficient	${∆}^{*}$	1.2
Initial gain coefficient	${∆}_{0}$	1.5
Maximum number of iterations		20
Predicted steps to control		15

**Table 4 biomimetics-08-00264-t004:** Reinforcement learning algorithm for dexterous hand manipulation.

01: Initialize these network parameters and DataBuffer, such as ${Dyn}_{\vartheta }$, ${Disc}_{\epsilon }$, ${Actor}_{\phi }$, ${Critic}_{\theta },{Value}_{\psi }$
02: Collect transfer data for the initial system state ⟶DataBuffer
03: Each episode executes in the iteration loop
04: Sample control targets randomly
05: The number of steps at each time t is executed in the iteration loop
06: Obtain $\left(s_{t},a_{t},s_{t+1},r_{t}\right)⟶DataBuffer,$ with the ${Actor}_{\phi }$ sampling motion
07: Policy updates in the iteration loop
08: Sample batch data from DataBuffer
09: Update the network parameters of ${Critic}_{\theta }$, ${Value}_{\psi }$, ${Actor}_{\phi }$, ${TargetValue}_{\overset{-}{\psi }}\left(s_{t}\right)$
10: Dynamic model updates in the iteration loop:
11: Sample batch data from DataBuffer
12: Update ${Dyn}_{\vartheta }$ and ${Disc}_{\epsilon }$
13: The HVAT sample is taken from the iteration loop
14: Sample the initial data from DataBuffer
15: The optimal trajectory planning is carried out to generate HVAT data ⟶DataBuffer

**Table 5 biomimetics-08-00264-t005:** Comparison of training sample efficiency of different algorithms.

Methodology	Reward-10,000	Reward-8000	Reward-6000	Reward-4000
Ours	6	10	20	25
SAC	7	12	21	30
PPO	135	151	162	N.A.
DDPG	7	18	63	N.A.

## Data Availability

There are no data to be shared.

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
