# Peer review of "A High-Efficient Reinforcement Learning Approach for Dexterous Manipulation"

_biomimetics, 2023, doi:10.3390/biomimetics8020264_

Round 1

Reviewer 1 Report

Major comments

1. The overall architecture is unclear. The proposed method consists of (1) modified Generative Adversarial Networks (GANs), (2) Differential Dynamic Programming (DDP), and (3) Soft Actor-Critic (SAC). The central difficulty in understanding is that the relation between the three components is unclear. For example, the authors did not explain how a training dataset is prepared for training the generator and the discriminator of (1). In Line 368, the authors mentioned that the SAC algorithm was employed to explore the environment, but the SAC algorithm (Section 2.5) needs HVAT data generated by trajectory planning. The trajectory planning (Sections 2.2 and 2.4) requires the model estimated by GANs. If the three components run simultaneously, I am unsure whether the overall algorithm is stable. Please clarify how the three components are organized.

Section 2.1: I have several questions described below:

2. Equation (1): The authors introduce the functions f and g, but the proposed method does not use the observation equation g. It would be better to remove the function.

3. The authors adopt the GANs framework for training the deterministic dynamical model. I am unsure whether the GANs framework is appropriate. If the authors adopt only the second term of the loss function (3), the learning process becomes more stable, and the entire architecture is straightforward.

4. Next, I do not understand the relation between the two loss functions because they differ from the standard ones. For example, the loss function of the discriminator should have log(D(s)) and log(1 -D(G(s))), where D and G represent a discriminator and a generator, respectively. However, the proposed method has log(s^), s^=Dyn(s, a)+s. The role of the discriminator is to distinguish the true s_(t+1) or generated s^_(t+1). What matters if the authors adopt the standard loss functions?

5. Section 2.2: Moving most parts to the appendix would simplify the manuscript because this section just explains differential dynamic programming. In addition, the appropriate references are needed.

6. Section 2.3: The right-hand side of Equation (28) is not a function of s. What does e mean? In addition, please add a line when alpha_s = 0.2 because it is used.

7. Section 2.5: Since the SAC algorithm is an entropy-regularized reinforcement learning algorithm, its objective function differs from that of DDP due to the entropy term. In addition, the SAC considers the discounted sum, while DDPP considers the finite sum. Would you discuss this mismatch?

8. In addition, there are several undefined functions in this subsection. For example, \tau_Buffer, EUCTN_t^(Dyn_\vartheta), Q^(s_t, a_t) in Equation (34) are not defined.

9. I do not fully understand the term (1/2)*(E[Critic(s, a’) + Critic(s, a)]) of Equation (35).

Section 3.1

10. Equation (46): The first equation seems strange. Is "-" be "+"?

11. Lines 381-382: The red curve should be the blue curve and vice versa.

Minor comments

Line 45: Yoshihisa -> Tsurumine

I did not find the reference to Figure 2.

Lines 141-142: \theta should be \vartheta, and \varepsilon should be \epsilon.

Line 166:  should be

Equation (5): \mu_i is not defined, although \mu is given in the main text. 

Author Response

The authors would like to express their sincere appreciation to the reviewer for his/her constructive comments and suggestions, and his/her time and efforts spent in helping us to improve the quality and presentation of the paper.

Reviewer 2 Report

It is Interesting and innovative work. 

The authors proposed a dynamic model of dexterous hands based on generative adversarial architecture, which is capable of learning the dynamic change principle of the dexterous hand system state. To solve the problem of prediction error explosion over long spans, the generative adversarial architecture was employed to learn the entire change rule of the dexterous hand system state while maintaining local prediction accuracy. ]

In my opinion, the work only requires a better presentation of the results and better formatting of the content. For example, the results presented in Figure 9 related to the prediction processes are not very clear. The same remark applies to the presentation of the results of evolution of the dynamic model's prediction error (Figure 10). The Reinforcement learning algorithm for dexterous hand manipulation shoul be formatted in algorithm LaTeX format. The key to the work diagram presenting the laboratory kit ( Figure 11) should be enlarged and better presented to the readers of the work. Table 5 is also not well formatted.

In my opinion, the work is interesting and after minor corrections it can be accepted for publication.

Author Response

(The authors gave the same response as above.)

Round 2

Reviewer 1 Report

The manuscript is revised well, but some reference information seem incorrect. I think they should be corrected and revised before acceptance. 

Author Response

The authors would like to express their sincere appreciation to the reviewer for his/her constructive comments and suggestions, and his/her time and efforts spent in helping us to improve the quality and presentation of the paper.

  • Comment: The manuscript is revised well, but some reference information seem incorrect. I think they should be corrected and revised before acceptance.

Response: Thank you for your helpful comment and kind suggestion. Reference information is modified according to the format of the journal as the attachment.
